# Life Cycle Assessment of the Gasoline Supply Chain in Sri Lanka

**Madhurika Geethani [1] and Asela Kulatunga [2],***

1   Department of Chemical and Process Engineering, Faculty of Engineering, University of Moratuwa, Moratuwa 10400, Sri Lanka; madhurikag@uom.lk
2   Department of Engineering, Faculty of Environment, Science and Economics, University of Exeter, Exeter EX4 4QF, UK
*   Correspondence: a.k.kulatunga@exeter.ac.uk

**Abstract:** The Sri Lankan transport sector still depends predominantly on petroleum fuels, mainly diesel and gasoline. Gasoline holds the second highest market share, and with the increasing number of gasoline-fueled vehicles, its proportion in the transport fuel mix is continuously expanding. The main objective of this study is to assess the ecological burden associated with the gasoline supply chain in Sri Lanka by conducting a life cycle assessment from a 'well-to-tank' perspective. In the scenario analysis, the environmental impacts of four potential gasoline distribution scenarios were assessed and compared with the existing distribution model. According to the results, the refining process was predominant, contributing more than 50% to climate change, terrestrial acidification, marine and freshwater eutrophication, human toxicity, and terrestrial and marine ecotoxicities. Meanwhile, crude oil extraction dominates in its contribution to ozone depletion, photochemical oxidant formation, freshwater ecotoxicity, and fossil depletion. The results of the scenario analysis show a remarkable reduction in the environmental load when rail transport is solely used to transfer gasoline from bulk terminals to regional depots. The reduction is over 65% in most impact categories compared to the existing distribution method, which involves a combination of both road and rail transport. This study identifies the key areas that need to be further analyzed to lower the environmental impacts while also establishing a foundation for conducting comparative environmental assessments of alternative fuel options in the Sri Lankan context.

**Keywords:** life cycle assessment; gasoline supply chain; environment; impact categories; Sri Lanka

## 1. Introduction

The primary energy demand of Sri Lanka comes mainly from petroleum and biomass, while the remainder stems from coal, hydropower, and other renewable sources. Among the two key players, petroleum dominates the country's energy mix, and that share has varied between 40 and 44% in the last few years [1–3]. Though energy end users such as household, commercial, and industrial sectors satisfy a significant portion of their energy requirement through renewable energy sources, the Sri Lankan transport sector still relies primarily on petroleum fuels. Diesel is the dominant fuel in the automobile market, while gasoline comes second. However, the annual gasoline demand has risen significantly over the last few years [4]. Worldwide also, there is a movement towards spark ignition engines, and the same is valid for Sri Lanka as the number of hybrid vehicles has increased due to the high fuel economy. On the other hand, there is a clear shift from public to private transport in Sri Lanka due to the various factors that are encouraging the local community to use their own modes of transportation [5]. Most of these private vehicles are fueled with gasoline; hence, the rise in numbers directly causes a boost in gasoline demand within the country.

The trend of rising gasoline demand is not limited to the Sri Lankan context; similar trends can also be observed in some other developing Asian countries such as Nepal, Myanmar and Cambodia. Gasoline consumption in Nepal has almost doubled during

the last five years, increasing the country's expenditure on imports. Furthermore, both Myanmar and Cambodia have witnessed a continuously increasing pattern in gasoline demand over the past years, despite the disturbances caused by COVID-19.

However, using petroleum fuels, including gasoline, is consistently linked to adverse effects on ecological systems. Air pollutants emitted from the combustion of petroleum fuels are a widespread concern that has become a prominent topic discussed in many environmental forums. In addition to tailpipe emissions, many activities related to fuel production, storage, and transportation negatively affect different components of the environment [6]. Consequently, critical evaluation of environmental impacts associated with every single supply chain step and taking necessary action is vital to eliminate or mitigate those harmful influences.

As a party to the Paris Agreement, Sri Lanka is bound to address environmental concerns and reduce the negative impact on the environment by eliminating or mitigating detrimental environmental activities. In September 2021, Sri Lanka submitted its updated nationally determined contributions (NDCs), including efforts to reduce national emissions [7]. Since the transport sector is a targeted sector in this journey, assessing the environmental profile of Sri Lankan fuel systems has also become necessary. However, a proper environmental assessment for the supply chain of petroleum fuels, including gasoline, still needs to be made available in the Sri Lankan context. Due to the significant rise of gasoline demand in the transport sector and as a starting point for assessing the environmental impact of petroleum fuels, the valuation of harmful influences associated with different stages of the gasoline supply chain is carried out through this study.

Life cycle assessment is a proven methodology for quantifying the environmental profile of a particular product or service. A number of LCA studies have been carried out in various countries to evaluate the ecological effect of the production, distribution, and usage of petroleum products [6–10]. Among LCAs conducted for petroleum fuels, only a few studies have focused solely on gasoline. Restianti and Gheewala [6] and Morales et al. [10] analyzed the environmental impact associated with the gasoline supply chain in their regions by conducting LCAs according to a well-to-wheel perspective. In both studies, the supply chain of gasoline was divided into four to five subsections, and the environmental impact associated with each subsection was assessed using a selected set of impact categories.

In this study, the Sri Lankan gasoline supply chain spread through well-to-tank was divided into seven subsections to focus more specifically on processes such as importation, distribution, and refueling as individual environmental impacts for these processes are limitedly reported. In addition, the fuel distribution process was comprehensively analyzed in this study by assessing different distribution scenarios. Accordingly, an environmentally favorable distribution method to transport gasoline from the primary storage terminals to regional depots was suggested through the scenario analysis, as the prevailing method cannot be considered optimal.

The rest of the paper is structured as follows: Section 2 describes the methodology used for conducting the LCA of the gasoline supply chain, as well as scenario analyses for fuel distribution. Section 3 presents and discusses the results of this study, while Section 4 contains the avenues for future research, the contribution of this study, and the conclusion of the paper.

## 2. Materials and Methods

The main objective of this study was to assess the environmental impacts of the gasoline supply chain in Sri Lanka using the LCA methodology. This study followed the International Standards Organization (ISO) 14040 and 14044 life cycle assessment standards [11,12]. After the completion of the main LCA, activities performed inside Sri Lanka were extracted into a separate LCA to highlight the environmental hotspots in the Sri Lankan operation.

## 2.1. System Boundary

The system boundary of the main LCA includes extraction of crude oil, storage and transportation of crude oil, the refining process, storage and distribution of gasoline, and the refueling process. The system boundary of the LCA conducted for the local operation includes all activities in the local gasoline supply chain up to the refueling stage. However, construction of infrastructure facilities and machinery was excluded from this study. Figure 1 presents an overview of the Sri Lankan gasoline system under study. In these analyses, the gasoline supply chain was divided into several subsystems; all activities contained in each subsystem are described in Table 1.

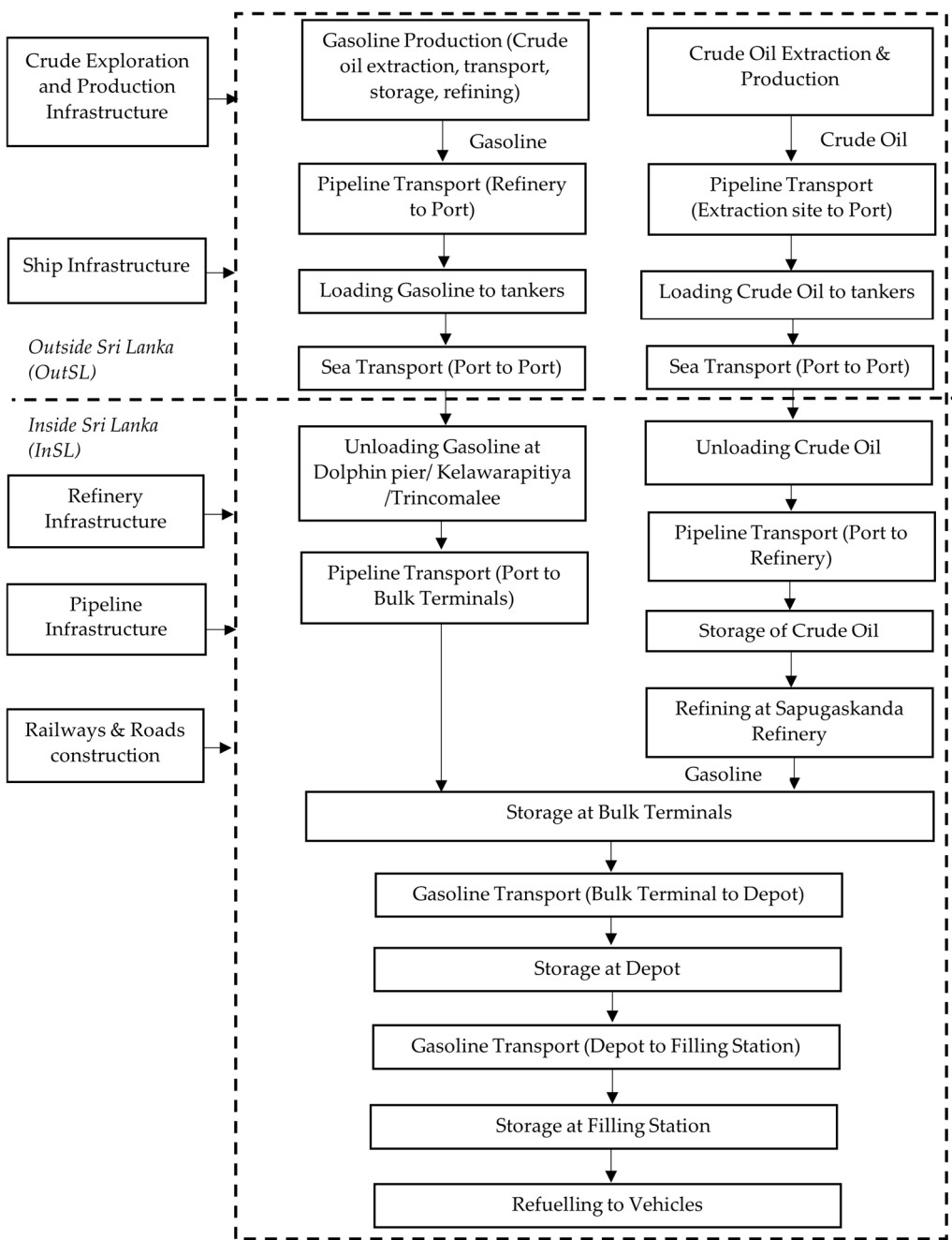

**Figure 1.** System boundary for the gasoline supply chain.

**Table 1.** List of activities included in subsystems.

| Subsystem | Description of Activities |
|---|---|
| Crude oil extraction | crude oil extraction, upstream processing activities, transportation inside the country of origin |
| Crude oil importation | crude oil loading to tankers, sea transport, unloading and ballasting activities, pipeline transport from port to storage |
| Gasoline production (local and imported) | crude oil storage inside the refinery, all activities associated with the refining process |
| Gasoline importation | loading refined gasoline into the tankers, sea transport, unloading, and ballasting activities, pipeline transport from port to bulk storage terminals |
| Gasoline storage | filling the storage tanks, storing in the tanks, pipeline transportation of gasoline inside the storage premises, emptying the tanks, loading, and unloading activities (gasoline storage in bulk terminals, regional depots, and service stations are considered here) |
| Gasoline distribution | transportation of gasoline from bulk terminals to regional depots, from regional depots to filling stations, direct transportation from bulk terminals to filling stations |
| Gasoline refueling | refueling of gasoline to vehicle tanks |

### 2.2. Functional Unit

In this study, one liter (1 L) of gasoline was used as the functional unit for normalizing input and output data. Since liters are the commonly used measuring unit for gasoline, this selection would allow other researchers to conveniently utilize the outcomes of this research in their own studies.

### 2.3. Allocation Procedure

Since the refinery provides multiple, strongly correlated products, partitioning of input and output flows of such processes is required to determine the appropriate share of the considered product. Although ISO standards recommend avoiding allocation whenever possible by dividing the unit process or expanding the product system, those approaches are not always applicable to processes such as refining. Therefore, as recommended, allocation can be performed based on the physical relationships of the products. Allocation in similar studies is commonly based on mass, volume, energy content, and market value shares. Among these different allocation methods, the market-value-based allocation approach provides an economic perspective, which could vary based on the period considered for data collection, while other methods provide an engineering perspective. Accordingly, mass-, volume-, and energy-based approaches were considered in selecting an appropriate allocation for this study. With the type of available data, mass- and energy-based allocations were preferred in the selection. However, a comparison conducted by Wang et al. [13] has revealed similar results in both mass- and energy-based allocations when they were used to allocate greenhouse gas emissions and energy usage to refinery products. In line with this, since the data sources most used in this study contain mass-related data, mass-based allocation was used to allocate energy use and emissions of the refinery process to refinery products.

### 2.4. Life Cycle Inventory

Primary data for this study were obtained mainly from annual, progress and audit reports issued by various governmental authorities and private sector stakeholders considering the time frame 2018–2019 [1,4,14–18]. When details were gathered to estimate LCI data, priority was given to 2018. As per observations, domestic transport activities were affected by the Sunday Easter attack in 2019 and then severely disturbed in 2020 and 2021 due to the impact of COVID-19 [4,19,20]. Therefore, the year 2018 was selected for this study as it reflects stable and undisturbed conditions in the gasoline supply chain.

However, there can be potential limitations associated with the use of 2018 data. Although operations of the gasoline supply chain were interrupted during the COVID-19

pandemic and the economic crisis peaked in 2022, most of these temporary disturbances have now settled in Sri Lanka, returning to a condition similar to that of 2018. Therefore, the impact remaining on the current gasoline supply chain due to COVID-19 and the economic crisis can be considered as minimal. However, over time, there could have been slight deviations in refinery operations, demand distributions, and technologies associated with different supply chain stages.

After the extraction of primary data, the remaining gaps were filled with secondary data extracted from the Ecoinvent 3 database and other existing literature. Key characteristics of the gasoline fuel supply chain and the main assumptions made during this study are described briefly below.

The Sapugaskanda refinery mainly processed Murban Crude (over 90%) during the considered period; therefore, it was assumed that the total refining process was performed for Murban crude [1]. Generally, this crude type is imported from the United Arab Emirates (UAE). Since specific inventory data for the considered region are not available, the data for crude oil extraction were obtained from the Ecoinvent database considering a world average. Imported crude oil is unloaded to Single Point Buoy Mooring 01 (SPBM 01) in Colombo harbor and transported to the Orugodawatta tank farm and then to the refinery [16]. The quantity of crude oil imported to Sri Lanka per functional unit of gasoline is included in Table 2.

**Table 2.** Quantity of crude oil and gasoline transported by pipelines (per functional unit).

| Pipeline Description | Transported Material | Volume (L) |
|---|---|---|
| SPBM located in the middle of the sea to Sapugaskanda refinery | Crude oil | 0.1342 |
| Colombo port to Kolonnawa terminal | Imported gasoline | 0.3870 |
| SPBM located in the middle of the sea to Muthurajawela terminal | Imported gasoline | 0.4558 |
| Sapugaskanda refinery to Kolonnawa terminal | Locally refined gasoline | 0.1072 |
| Sapugaskanda refinery to Sapugaskanda terminal | Locally refined gasoline | 0.0331 |
| China Bay harbor to China Bay tank farm (Trincomalee tank farm) | Imported gasoline | 0.0169 |

In 2018, 86% of Sri Lanka's gasoline demand was fulfilled through imports, while the refinery provided the rest of the requirement [4]. Gasoline and other petroleum fuels are imported from several countries, including Singapore, the UAE, Malaysia, and India. The proportion of imports from each country was determined based on the total imported tonnage. The countries that supply refined gasoline in small quantities (less than 5% of the whole imported tonnage) were not considered while modeling the gasoline importation subsystem [21,22].

Tankers with refined gasoline reach Dolphin Pier in Colombo harbor, SPBM 02 in Kerawalapitiya, and Trincomalee unloading points [16]. The quantities of gasoline unloaded at each site per considered functional unit are included in Table 2.

Gasoline unloaded in Dolphin Pier, SPBM 02, and Trincomalee is transferred to Kolonnawa, Muthurajawela, and Trincomalee tank farms using pipelines [18]. For both gasoline and crude importation, the Ecoinvent database was used in modeling sea transport and pipeline transport, while energy consumption and air emissions associated with loading and unloading activities were determined based on the literature [23,24]. The refining process for imported gasoline was modeled using the Ecoinvent database, while inventory data for the local refining process were developed using the information obtained from governmental reports [14,16]. With the collected refinery input and output data, the mass and energy balance closure were checked for the Sapugaskanda refinery operation. Some data that were lacking, such as cooling water usage and refinery heat loss, were determined using the details extracted from the literature [25,26]. All emission terms were determined based on refinery norms and standards. Accordingly, air emissions from the refining process were assumed to stem mainly from combustion, process, and fugitive emissions. Fuel combustion emissions were modeled using the Ecoinvent database, and other process emissions were based on the AP-42 fifth edition and other relevant literature

sources [23–25]. The life cycle inventory for producing 1 L of gasoline in the Sapugaskanda refinery is shown in Table 3. Here, all inputs and outputs of gasoline production have been tabulated after performing mass-based allocation.

**Table 3.** Life cycle inventory for refining 1 L of gasoline.

| Description | Value | Unit |
|---|---|---|
| **Inputs** | | |
| *Materials* | | |
| Crude oil | $7.882 \times 10^{-1}$ | kg |
| Water (from river) | $7.749 \times 10^{-2}$ | kg |
| Water (from WB) | $3.611 \times 10^{-1}$ | kg |
| *Energy* | | |
| Fuel oil * | $9.473 \times 10^{-1}$ | MJ |
| Fuel gas (Refinery gas) * | 1.112 | MJ |
| Electricity * | $6.170 \times 10^{-3}$ | MJ |
| **Outputs** | | |
| *Product* | | |
| Gasoline | 1.000 | L |
| *Air emissions (without fuel combustion and electricity-related emissions)* | | |
| CO | $1.148 \times 10^{-2}$ | g |
| $NO_X$ | $5.750 \times 10^{-2}$ | g |
| PM10 | $5.629 \times 10^{-1}$ | g |
| $SO_2$ | $1.157 \times 10^{-1}$ | g |
| $CO_2$ | 6.194 | g |
| Hydrocarbons | 1.751 | g |
| Aldehydes | $2.873 \times 10^{-3}$ | g |
| Ammonia | $8.250 \times 10^{-3}$ | g |
| Water vapor | $3.259 \times 10^{-1}$ | kg |
| *Soil emissions* | | |
| Oil spills | $3.826 \times 10^{-1}$ | g |
| *Water emissions* | | |
| Wastewater | $1.127 \times 10^{-1}$ | kg |
| *Solid wastes to treatment* | | |
| Solid wastes | 4.578 | g |

* Fuel oil and fuel gas combustion and electricity generation were modeled using Ecoinvent databases. Emissions generated from those processes are not included here.

Both imported and refined gasoline are initially stored in bulk terminals, then in regional depots, and finally in underground filling station tanks. Eleven regional bulk stations are available, and all these depots, excluding Sarasavi Uyana, are used to store gasoline [16]. Emissions and energy requirements per functional unit of gasoline stored in a bulk terminal and a regional depot were taken as the same, considering the similarities in functions.

Gasoline is conveyed between bulk terminals and regional depots using road bowsers and rail fuel tankers [18,27,28]. It was assumed that direct purchasing of fuels is only performed by dealers in Colombo, Gampaha, Kalutara, Puttalam, and Ratnapura districts and that the CPSTL delivers gasoline directly to the abovementioned districts using road transport, as those areas are closer to bulk terminals than to regional depots. When modeling the road haulage between the depot and filling stations in the rest of the districts, it was assumed that the fuel demand of a particular district is satisfied from the nearest regional depot. Ecoinvent databases were used to model the fuel's road and railway transport. Furthermore, the amount of gasoline unloaded to the Trincomalee tank farm is assumed to be only delivered to filling stations in the Northern, North Central, and Eastern

provinces. Since the loading and unloading of trucks and railway wagons are performed inside the storage premises, those emissions were also included in the storage subsystem. As the final stage of the considered boundary, the refueling consumes energy, generates VOC emissions, and causes spills. Based on the flow rate of a standard duty dispenser generally used in local filling stations, the power requirement to refuel 1 L of gasoline was determined. All air emission values related to gasoline storage (filling, emptying, and breathing), loading and unloading activities, and the refilling process were determined based on the information obtained from the literature [23,24].

### *2.5. Scenario Analysis*

Different options that can be used to deliver 1 L of gasoline from bulk terminals to regional depots were evaluated during the scenario analysis without considering other life cycle stages. The existing condition is considered as the base case scenario. After performing separate LCA analyses for scenarios 1, 2, 3, and 4, the obtained results were compared with the outcomes of the base case scenario.

In addition to the comparative environmental impact assessment, an operational cost comparison of four scenarios was conducted to gain an understanding of the economic implications of each scenario. Fuel transportation costs for upcountry and low-country routes were extracted from the literature in line with the timeframe considered in the LCI development.

### 2.5.1. Scenario 01 (S01)

In the first scenario, gasoline is transported from bulk terminals to regional depots using rail fuel tankers only. Freight transportation using railways has proven to be environmentally sustainable, even though railway tankers are powered by diesel engines. Previous work carried out on petroleum distribution using railway wagons in Sri Lanka has also proven the cost-effectiveness of using railways. However, fuel distribution in Sri Lanka still relies primarily on road transportation. Therefore, this scenario quantifies the environmental benefit that can be achieved if fuel transportation between bulk terminals and regional depots is completely moved to railway transport.

As the Muthurajawela terminal does not contain railway infrastructures, either a railway facility should be constructed in the Muthurajawela terminal or an interconnecting pipeline should be laid between the two main terminals to transport fuel from Muthurajawela to Kolonnawa. Since a feasibility study has already been carried out to construct pipelines connecting the two terminals, that alternative is considered here.

### 2.5.2. Scenario 02 (S02)

In the second scenario, gasoline transportation between bulk terminals and regional depots is wholly carried out through road tankers. As mentioned in the first scenario, road transport has received priority in fuel distribution in Sri Lanka, primarily due to the lack of railway wagons. The second scenario analyzes how much burden will be placed on the environment if fuel transportation is gradually shifted towards a complete dominance of road transportation.

### 2.5.3. Scenario 03 (S03)

The optimum distribution model proposed by Gunaruwan and Sannasooriya [27] is applied to distribute gasoline to regional depots in the third scenario. The lack of railway wagons has been identified as the primary reason for the poor utilization of railway wagons in fuel distribution. Gunaruwan and Sannasooriya [27] suggested an optimized model in which existing railway wagons are prioritized for flat terrain fuel distribution, while road bowsers are only used for upcountry destinations. This optimized model has demonstrated its capability for significantly reducing fuel consumption. The third scenario evaluates the potential reduction in different environmental impact categories when this model is implemented.

Accordingly, all up-country destinations (Peradeniya, Kotagala, Haputale, Badulla) are to be reached entirely via road bowsers, while destinations at further distances over flat terrain (Matara, Galle, Kurunegala, Anuradhapura, Batticaloa) are to be connected to the bulk terminal through rail fuel tankers. However, since railway facilities are not available up to the Kankesanthurai (KKS) depot as those infrastructures were destroyed during the war, replenishment of the KKS depot is to be carried out entirely by road bowsers.

### 2.5.4. Scenario 04 (S04)

Supplying gasoline to regional depots from the nearest storage terminal was the amendment to the base case scenario in developing the fourth scenario. Accordingly, depots in Anuradhapura, Batticaloa, and KKS are to be fed from the China Bay tank farm in Trincomalee while the delivery of fuels to other depots is carried out through the Muthurajawela and Kolonnawa terminals, just as in the base case scenario.

The China Bay tank farm has been underutilized over the past decades, disregarding its vast storage capacity and other associated benefits such as its location adjacent to the Trincomalee harbor and integration with the previously established railway system. As per the previous inspections carried out in the tank farm, a certain number of the idling storage tanks can be reused after minor renovations. Accordingly, instead of transporting fuel from Colombo to Trincomalee and surrounding areas, replenishing fuel requirements in those areas from the Trincomalee tank farm is a proven, economically viable option. The fourth scenario was chosen to quantify the environmental productivity of supplying gasoline from the China Bay tank farm to nearby depots while others are fed as usual. Here, it was considered that the fuel transportation from the China Bay terminal to Anuradhapura, Batticaloa, and KKS is carried out entirely by rail fuel tankers.

### 2.6. Modeling and Life Cycle Impact Assessment

The SimaPro 8.3.0.0 software package was used to model the LCAs and perform the impact assessment analysis. With the use of collected data and Ecoinvent databases included in the software package, the main life cycle assessment on the gasoline supply chain was initially modeled as an integration of seven subsystems as defined in Table 1.

When Ecoinvent databases were used to model the refining process of imported gasoline, fuel combustion in the local refining process, and transport activities, Rest of World (RoW) databases were used as country-specific databases were not available. However, necessary modifications to those databases were made, if required, considering the unique characteristics of local operations.

Then, an impact assessment was conducted using the ReCiPe V1.13 midpoint method in the hierarchic perspective. The midpoint characterization has a stronger connection to the environmental flows and a relatively lower uncertainty compared to the endpoint characterization. Since the midpoint approach transforms the LCI results into various impact categories, it provides information about the amounts of inputs and outputs that can potentially contribute to a specific environmental concern. In addition, the ReCiPe framework includes a broad set of updated impact categories and is considered as an up-to-date methodology. Considering all these facts, the ReCiPe V1.13 midpoint method was used to conduct LCIAs in this study. The results of twelve selected environmental impact categories were considered during this study.

Thereafter, local activities in the gasoline supply chain were extracted into another LCA and analyzed separately to identify the environmental hotspots present in local operations. Finally, four scenarios were modeled considering their specific distribution characteristics, and a comparative impact assessment was performed for all scenarios and the existing distribution mechanism.

## 3. Results and Discussion

Twelve impact categories including climate change (CC), ozone depletion (OD), human toxicity (HT), photochemical oxidation formation (POF), terrestrial acidification (TA),

freshwater eutrophication (FE), marine eutrophication (ME), terrestrial ecotoxicity (TET), freshwater ecotoxicity (FET), marine ecotoxicity (MET), urban land occupation (ULO), and fossil depletion (FD) were considered during the impact assessment of this study. Figure 2 shows the contribution to the impact categories from each subsystem of the supply chain. According to the results, crude oil extraction and gasoline production (refining process) played significant roles compared to other subsystems. A detailed description of each impact category is given below. In the description, the refining of imported and local gasoline is considered as one subsystem, and in Figure 2, these processes are shown separately to illustrate the contribution of each process.

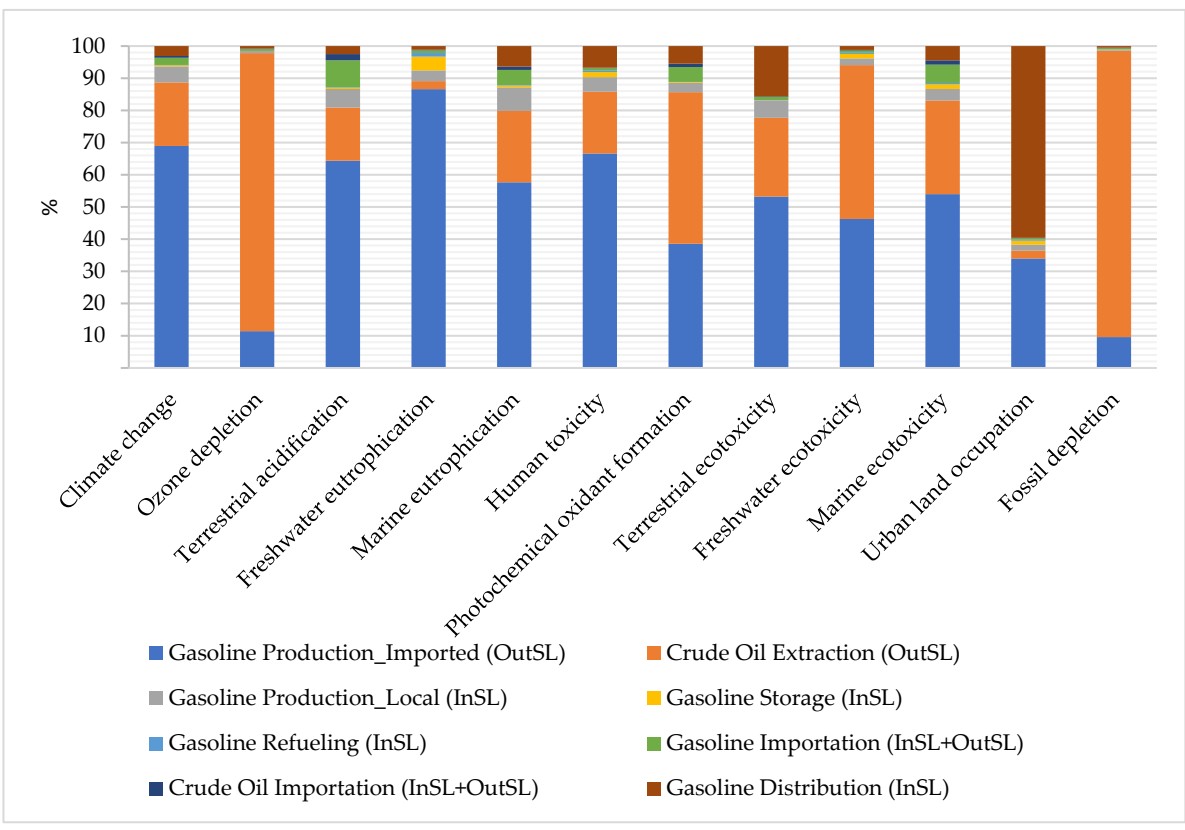

**Figure 2.** Contributions of subsystems in the gasoline supply chain to the various impact categories.

### 3.1. Climate Change

The emission of greenhouse gases per functional unit is 0.45 kg $CO_2$ eq. Refining gasoline is the primary process contributing to climate change, accounting for 74% of the total impact, followed by crude oil extraction (20%). A collective influence of refinery gas and heavy fuel oil combustion has caused the refining process to become the most dominant contributor in this category. Refinery gas combustion contributes 51% of the total refinery GHG emissions, while fuel oil combustion generates 27% due to the high proportion of $CO_2$ released from these activities.

### 3.2. Ozone Depletion

The value of OD per functional unit is estimated as $3.30 \times 10^{-7}$ kg CFC-11 eq, with the hotspots being crude oil extraction with an almost 86% contribution, followed by the refining process with around 12%. The emission of VOCs, flaring and venting activities, and chemical usage involved in the crude oil extraction process are the main causes of the recorded high contribution to OD.

*3.3. Terrestrial Acidification*

The total TA per functional unit is $2.98 \times 10^{-3}$ kg $SO_2$ eq, which is shared primarily by three main process components—refining of gasoline (70%), crude oil extraction (17%), and gasoline importation (8%). Heavy fuel oil burning for energy production contributes approximately 50% of the TA generated from refining activities.

*3.4. Eutrophication Potentials*

This study analyzed two eutrophication potentials: freshwater eutrophication (FE) and marine eutrophication (ME). Concerning FE, most of the impact of $2.36 \times 10^{-5}$ kg P eq. is from the refining-related activities of gasoline (90%) mainly due to the emissions associated with electricity production. The contribution of electricity-related emissions accounts for 62% of the FE caused by refining activities. The same trend applies to ME, where the refining processes contribute to 65% of the total ME of $6.65 \times 10^{-5}$ N eq. Emissions generated from processing activities and fuel combustion in refineries contribute primarily to this high-impact ME. Gasoline storage (4%) is the second largest contributor to FE, while crude oil extraction (22%) is the next largest contributor to ME.

*3.5. Human Toxicity*

As per the analysis, the total HT value per functional unit is $4.02 \times 10^{-2}$ kg 1,4-DB eq, predominantly driven by the contributions of refining activities (71%) and the crude oil extraction process (19%). Emissions generated from fuel oil combustion in the refining process are the main contributing factor (62%) for HT related to refineries.

*3.6. Photochemical Oxidant Formation*

The POF impact ($2.15 \times 10^{-3}$ kg NMVOC) is affected mainly by emissions derived from the refining activities and crude oil production processes. Among those two, crude oil extraction plays the leading role, contributing 48%, while refining imported and local gasoline collectively accounts for 41%.

*3.7. Ecotoxicity Potentials*

During this study, three impact categories related to ecotoxicity were assessed: terrestrial ecotoxicity (TET), freshwater ecotoxicity (FET), and marine ecotoxicity (MET). The total TET per functional unit is $3.35 \times 10^{-5}$ kg 1,4-DB eq, and the estimated value for MET is $1.01 \times 10^{-3}$ kg 1,4-DB eq. The refining process significantly contributes 59% and 58% to TET and MET, respectively. Fuel oil combustion generates many pollutants, contributing 69% and 65% of TET and MET of the refining process. Crude oil extraction is the second largest contributor towards both of these categories. On the other hand, the total FET of $1.09 \times 10^{-3}$ kg 1,4-DB eq is equally shared between the same two process components with a proportion of 48%.

*3.8. Urban Land Occupation*

The total ULO per functional unit is $7.30 \times 10^{-4}$ m$^2$a, in which refining crude oil and gasoline distribution play significant roles. Gasoline distribution is responsible for almost 60% of the impact, followed by the refining process (36%). As the moving of fuel carriers occupies a considerable amount of urban area for a considerable period of time, gasoline distribution contributes significantly to ULO.

*3.9. Fossil Depletion*

The amount of fossil depletion per functional unit is 0.97 kg oil eq, which depends dominantly on crude oil extraction with a contribution of 89%. This is due to the crude oil being extracted from nature. The next contributor is the refining process, with a contribution of 10%, due to fossil fuel consumption as energy sources during refinery activities.

According to the LCA results, the environmental burden associated with the gasoline supply chain can be reduced primarily through the refining process. In the refining process,

energy production has a resultant higher impact value in most impact categories due to the massive amount of air pollutants that are released into the environment during the combustion of energy sources, including fuel gas and fuel oil. Approximately 80% of the total energy requirement of refineries is generally satisfied internally through the burning of byproducts. Therefore, careful evaluation of the methods used to satisfy the energy requirements in refineries and resolving associated problems is a key step in addressing other remaining environmental concerns.

Besides the main environmental hotspots, gasoline importation and distribution have a noticeable impact on the environment. As mentioned in Section 2.4 the majority of the country's gasoline demand is satisfied through imports, exclusively relying on sea tankers as the mode of transport. Most marine vessels are powered by burning fuels which generally contain up to 3.5% sulfur and other incombustible materials. Hence, the operation of a marine engine results in emitting a number of harmful airborne particles, including a vast number of particulate matters which are the main cause of the environmental burden associated with importation activities [29].

As one solution, a multi-product pipeline from southern India to Sri Lanka has been proposed to ensure a reliable supply of energy sources [30]. This will significantly reduce the contribution of gasoline importation to impact categories such as CC, TA, ME, POF, and MET, in which the contributions of gasoline importation are high due to sea transport using tankers. Construction of the pipeline will also reduce the environmental impacts related to the distribution since it will guarantee a stable fuel supply to the Northern region. The gasoline distribution is analyzed and discussed under local operations and scenario analysis.

The results obtained from the LCA were compared with similar studies to provide an LCA perspective. Several previous studies have identified the refining process and fuel usage as the most environmentally destructive activities [6,8,10]. This study identified refining as the most crucial subsystem, followed by crude oil extraction. Accordingly, there is an agreement with the findings in the literature, as tailpipe emissions do not apply to our study.

Most research has been conducted within well-to-wheel system boundaries, although this work focused on the environmental aspects of a well-to-tank system [6,9,10,31]. Therefore, a comparison can only be made between the impact category results when the contribution of the usage phase (combustion stage) to a specific impact category is provided in the publication. The results of three studies performed by Morales et al. [10], González-García et al. [32], and Restianti and Gheewala [6] were used for the comparison. However, only the results of climate change and terrestrial acidification could be compared. Regarding CC, results in the literature range from 1.195 to 0.161 kg $CO_2$ eq, and the CC value obtained for the Sri Lankan context (0.45 kg $CO_2$ eq) lies within that range. Though the result in this study is lower than the CC value reported by Morales et al. [10] for the Chilean supply chain (1.195 kg $CO_2$ eq), it is slightly higher than the results of the other two studies. Similar to CC, the highest value for TA was reported by Morales et al. [10], and it was $4.284 \times 10^{-3}$ kg $SO_2$ eq. Our value ($2.985 \times 10^{-3}$ kg $SO_2$ eq) falls between the results obtained by González-García et al. [32] for the Spanish context ($3.258 \times 10^{-3}$ kg $SO_2$ eq) and Restianti and Gheewala [6] for the Indonesian context ($8.670 \times 10^{-4}$ kg $SO_2$ eq). It is noteworthy that there are remarkable differences in the results, even in relation to the same functional unit.

Compared to the characteristics of the Sri Lankan gasoline chain, the gasoline supply chains evaluated in both the Chilean and Indonesian studies exhibit more environmentally friendly features. In the Indonesian study, crude oil is transported to the refinery using a pipeline, and refined gasoline is transferred to the depot and subdepot using pipelines and trains, respectively. The length of these transportation routes can be lower compared to Sri Lanka as both extraction and refining are completed nationally. With these characteristics, the lower impact values reported by Restianti and Gheewala [6] can be supported.

A portion of crude oil in the Chilean study is also supplied through pipelines, while gasoline for storage plants located in different regions is transferred through pipelines.

Furthermore, their refining process is known as a complex and highly intertwined process, customized for specific customized properties. However, the impact values reported by Morales et al. [10] are higher than those in this study. In the Chilean study, the production of chemicals and machinery, construction of infrastructure, and maintenance activities were included in the system boundary. This inclusion, which is not applicable to our study, can be a reason for the higher impact values reported for CC and TA. In addition to the specific characteristics of the processes in the considered region and slight differences in system boundaries such as inclusion or exclusion of infrastructure activities, the sources of inventory data used, allocation approaches, and characterization techniques can also contribute to the variations among reported impact category results.

### 3.10. Local Operations of the Gasoline System

In the analysis conducted for Sri Lankan operations of the gasoline supply chain, the local refining process and gasoline distribution dominate in most of the impact category results. Though the environmental load generated from refining 1 L of gasoline is significantly higher than that for the distribution of 1 L, the fraction of locally refined gasoline in the considered functional unit is only around 14%. Therefore, the distribution process of gasoline carries a comparatively high environmental load in local operations. Fuel oil and refinery gas combustion in the refinery makes a significant contribution to the refining process, while emissions generated from the combustion of automobile fuels during transportation activities affect the environmental load of the gasoline distribution. The subsystem of gasoline distribution is a collection of different transportation activities; therefore, contributions of those to various impact categories have been assessed. Correspondingly, the transportation of gasoline from bulk terminals to regional depots is the most significant contributor of all considered impact categories, since it is the section that handles the largest quantity of gasoline over the farthest distance.

### 3.11. Scenario Analysis on Fuel Distribution

The comparison of the environmental impacts associated with each distribution model, analyzed during the scenario analysis, is shown in Figure 3. As this scenario analysis aims to quantify the environmental benefit or detriment associated with each of the selected scenarios compared to the base case, Table 4 has summarized those details. The change represents the impact of substituting the base case with any of the four alternative scenarios. A negative change indicates a decrease in the environmental load compared to the base case, and a positive value indicates an increase in the environmental load. In the column of the base case scenario, the annual environmental load of each category has been tabulated to obtain a clear overview of the quantified benefit or damage that can occur annually.

**Table 4.** Percentage of change in impact categories due to the substitution of the alternative scenarios for the base case.

| Impact Category | Annual Env. Load of Base Case Scenario | Scenario 01 % of Change | Scenario 02 % of Change | Scenario 03 % of Change | Scenario 04 % of Change |
|---|---|---|---|---|---|
| CC (kg $CO_2$ eq) | $1.20 \times 10^7$ | −67.5 | +45.1 | −34.7 | −20.5 |
| OD (kg CFC-11 eq) | 2.29 | −68.1 | +45.6 | −34.8 | −20.7 |
| TA (kg $SO_2$ eq) | $6.76 \times 10^4$ | −43.0 | +29.1 | −22.5 | −15.2 |
| FE (kg P eq) | $2.24 \times 10^2$ | −85.6 | +57.0 | −43.4 | −24.8 |
| ME (kg N eq) | $3.85 \times 10^3$ | −38.1 | +25.4 | −20.6 | −14.1 |
| HT (kg 1,4-DB eq) | $2.18 \times 10^6$ | −97.3 | +64.7 | −48.9 | −27.5 |
| POF (kg NMVOC) | $1.08 \times 10^5$ | −37.0 | +25.3 | −19.9 | −13.7 |
| TET (kg 1,4-DB eq) | $4.22 \times 10^3$ | −97.3 | +64.4 | −49.0 | −27.4 |
| FET (kg 1,4-DB eq) | $1.27 \times 10^4$ | −82.5 | +33.8 | −48.2 | −33.1 |
| MET (kg 1,4-DB eq) | $3.61 \times 10^4$ | −95.0 | +63.8 | −47.8 | −26.8 |
| ULO ($m^2$a) | $3.49 \times 10^5$ | −99.8 | +66.7 | −50.1 | −27.9 |
| FD (kg oil eq) | $4.28 \times 10^6$ | −68.3 | +45.2 | −35.0 | −20.8 |

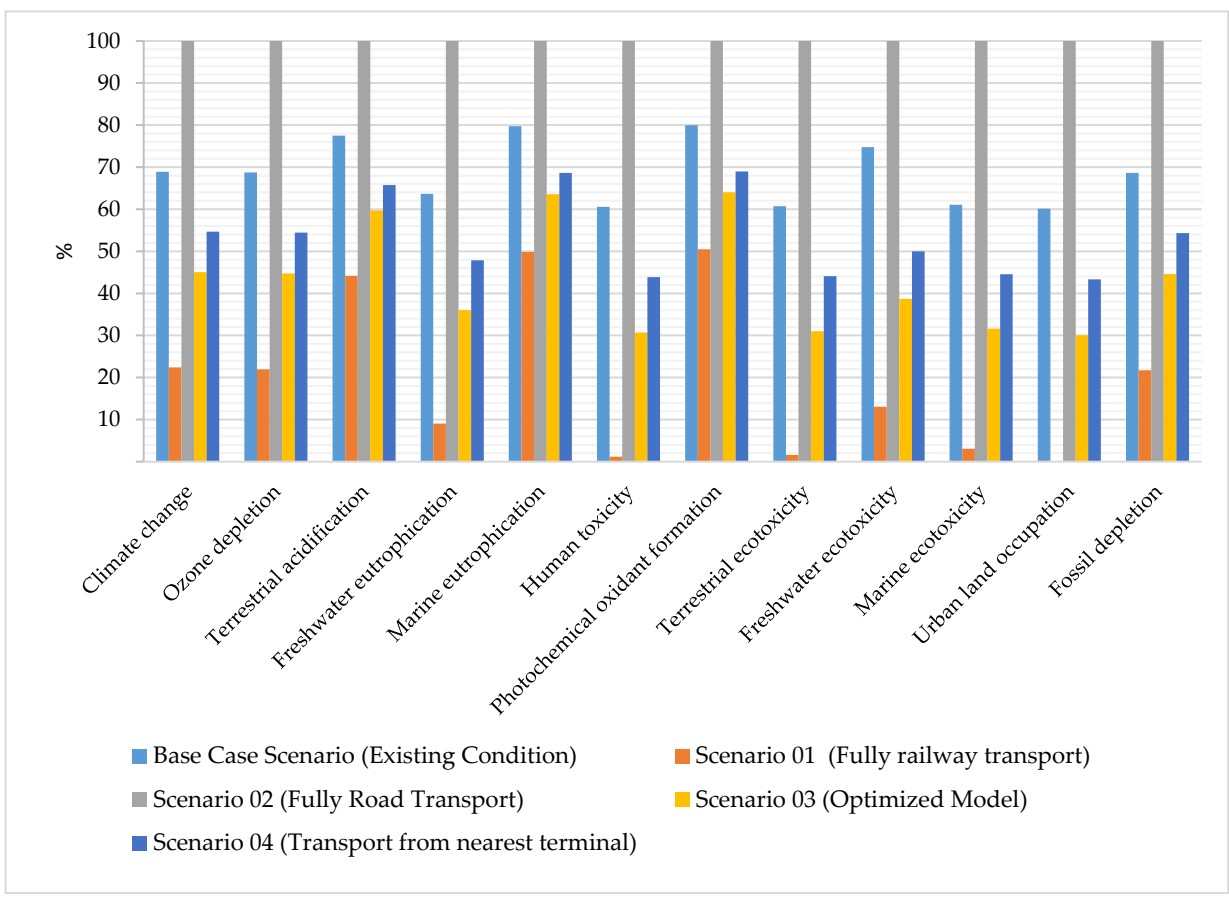

**Figure 3.** Comparison of environmental impacts of different scenarios of fuel distribution.

The highest impact on the environment is generated when gasoline is transported via road bowsers (S02), while the lowest impact on the environment is observed when fuel transportation is carried out through rail fuel tankers (S01). According to Table 4, a reduction of more than 60% can be achieved in the impact categories of CC, OD, FE, HT, TET, FET, MET, ULO, and FD when rail fuel tankers are used for gasoline transportation between bulk terminals and regional depots. In contrast, there is more than a 40% increase in most of the impact category results due to the usage of road bowsers. Several researchers have compared the environmental burden of different freight transport modes in various regions using the LCA methodology. In each study, rail transport has been evaluated as a freight transport mode, and the results have shown a significantly lower burden on the environment compared to road transport, even though the trains selected for this study were powered by diesel [33–35]. However, in this study, driving distances of rail and road between bulk terminals and regional depots are not similar. In most cases, rail transport distances exceed road driving distances. Regardless of higher driving distances, the scenario analysis proved that railway transportation of the fuel has a higher capacity to reduce the environmental impacts associated with fuel distribution.

For most impact categories, the optimized model (S03) proposed by Gunaruwan and Sannasooriya [27] showed better results than the base case, S02, and S04, and was only second to S01. In this third scenario, rail fuel tankers are mainly allocated to regional depots at greater distances over flat terrain (Matara, Galle, Kurunegala, Anuradhapura, and Batticaloa). Demand in these depots is also comparatively higher than that in the rest of the depots. Accordingly, in the optimized model, the proportion of gasoline transported via rail fuel tankers and the distance covered by the rail fuel tankers are greater than the amount of gasoline and transportation distance handled by the road bowsers. Consequently, the usage of the third scenario has shown a remarkable reduction in the environmental load.

The lack of rail fuel tankers has been identified as a key reason for the limited usage of railway transport for fuel distribution. The analysis shows that the third scenario is a viable solution for this concern, as environmental load reduction in the third scenario is observed by allocating fewer rail fuel tankers in a more efficient manner.

Although the environmental benefits of S04 are lower than those of S01 and S03, there is still more than a 20% reduction in all impact categories except TA and ME, compared to the base case scenario. Accordingly, replenishing the gasoline requirements of Anuradhapura, Batticaloa, and Kankesanthurai depots from the China Bay tank farm via railway fuel tankers can considerably decrease the environmental load.

The outcomes of the operational cost comparison of the four scenarios also exhibit the same pattern of variation. The annual operational cost incurred for transferring gasoline from bulk terminals to regional depots in BCS was estimated at approximately 1050 million Sri Lankan rupees. When gasoline transportation is carried out by rail, the potential reduction in annual operational cost is 57%. In contrast, a 36% increase in operational cost could be seen in the road transportation of gasoline. Both the third and fourth scenarios also showed a positive effect on the operational cost, reducing it by 28% and 17%, respectively. As the unit transport cost incurred for road bowsers is around three times higher than that incurred for railway wagons, a significant annual saving can be attained when the proportion of gasoline transported by rail is increased. According to the results, the first, third, and fourth scenarios are not only capable of decreasing environmental load but also have the potential to make significant annual operational cost savings. However, conducting a detailed economic cost analysis, including a life cycle costing, will be required to obtain a complete economic picture of these distribution scenarios.

Some infrastructure development related to the fourth scenario has already been started in Trincomalee. The development of the China Bay tank farm has been initiated; in addition to the 14 tanks operated by Lanka IOC (LIOC), another 85 tanks are being leased to Ceylon Petroleum Cooperation (CPC) and Trinco Petroleum Terminal (Pvt) Limited (TPTL) for a duration of fifty years. Accordingly, the leased tanks will be repaired, and CPC has already started to develop 12 tanks of the 24 tanks leased to them [36]. After completion of the development, the fuel requirement in the North, East, and North Central provinces will be completely satisfied through the Trincomalee tank farm. This study can be used as supporting evidence to highlight the possible environmental gains embedded in those new developments. Similarly, this study can serve as a baseline for conducting a comparative environmental assessment between the current condition and any modifications in the gasoline fuel supply chain. Moreover, the analysis could be extended to assess the environmental sustainability of future infrastructure developments related to fuel storage and distribution in Sri Lanka.

## 4. Conclusions

The paper provides the results of impact categories of the LCA performed on the gasoline supply chain in Sri Lanka. The principal analysis covered all the processes from crude oil extraction to the refueling stage. Crude oil extraction and refining processes were identified as the critical environmental hotspots within the scope considered. Of these two, the refining process is the most significant contributor towards most impact categories, including CC, TA, FE, ME, HT, TET, and MET, with ratios of more than 50%, while crude oil extraction dominantly affects the categories of OD, POF, FET, and FD. For the impact category of ULO, gasoline distribution is responsible for more than half of the impact value. When the local operations of the gasoline supply chain were analyzed separately, the local refining process and gasoline distribution were found to be vital environmental hotspots. The scenario analysis evaluated the environmental impacts of four different transportation options that can be used to deliver gasoline from bulk terminals to regional depots. Rail fuel tankers were identified as the most beneficial scenario for the environment, and the optimized model proposed by Gunaruwan and Sannasooriya [27] also shows a remarkable reduction in environmental loads.

Since this study has relied mainly on published data, there are limitations in LCI. Therefore, the quality of the work can be improved by using actual field data. In addition, a detailed economic cost analysis including a life cycle costing of the four distribution scenarios is required to obtain a proper understanding of the economic aspects associated with each scenario.

However, the outcomes of this study can be used as a basis for a comparative assessment of various fuel options. Furthermore, this study provides sufficient information regarding the main environmental hotspots of Sri Lankan operations, which will assist energy regulatory bodies in taking necessary action to reduce the environmental burden generated by the gasoline supply chain. The outcomes stemming from scenario analysis can also assist decision-makers in fuel distribution when selecting different distribution structures, while government authorities responsible for related infrastructure developments can utilize this information to prioritize their development activities.

**Author Contributions:** Conceptualization, M.G. and A.K.; methodology, M.G. and A.K.; software, M.G.; validation, M.G. and A.K.; formal analysis, M.G.; investigation, M.G. and A.K.; resources, A.K.; data curation, M.G.; writing—original draft preparation, M.G.; writing—review and editing, A.K.; visualization, M.G.; supervision, A.K.; project administration, A.K. All authors have read and agreed to the published version of the manuscript.

**Funding:** This research received no external funding.

**Institutional Review Board Statement:** Not applicable.

**Informed Consent Statement:** Not applicable.

**Data Availability Statement:** The data presented in this study are available on request from the corresponding author.

**Conflicts of Interest:** The authors declare no conflicts of interest.

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
