# Peer review of "Life Cycle Assessment of the Gasoline Supply Chain in Sri Lanka"

_sustainability, doi:10.3390/su162410933_

Round 1
Reviewer 1 Report
Comments and Suggestions for Authors
The article is interesting and relevant. Implementing the following recommendations could further imrpove it.
I. Introduction
While the authors highlight the rising gasoline demand in Sri Lanka, a brief comparison to trends in similar regions or developing nations would strengthen the study's relevance. Improving the flow by organizing information chronologically would enhance clarity: begin with the current energy scenario, proceed to environmental concerns and policy commitments, outline gaps in existing research, and conclude by stating the study’s objective.
II. Methods
Primary data were collected from government and industry reports from 2018. Detailing potential limitations of using data from 2018—especially considering recent developments or disruptions, such as COVID-19, which may have altered the gasoline supply chain—would add valuable context. Expanding on the rationale for selecting mass-based allocation over other methods (e.g., energy or market-based allocation) and discussing any resulting impacts would further strengthen the methodology. Additionally, a brief justification for employing the ReCiPe midpoint method for impact assessment—such as its applicability to relevant environmental impact categories in the gasoline supply chain—would enhance the section's robustness.
III. Results
The authors should clarify regional differences by emphasizing factors unique to Sri Lanka, such as infrastructure limitations or regional environmental policies. This would help clarify why the findings may diverge from other studies.
IV. Conclusion
Highlight how stakeholders (e.g., policymakers, fuel distributors) might leverage these findings to prioritize rail transport or optimize processes to reduce emissions in the concusion section.
Comments on the Quality of English LanguageThe English requires a minor review.
Reviewer 2 Report
Comments and Suggestions for Authors
Dear Autrhors,
The paper's title accurately reflects the paper's content, providing clarity on the study's scope and geographical scope (Sri Lanka). The objective of the study, to assess the environmental impacts of the gasoline supply chain from a life cycle perspective, is well-articulated and relates to current discussions on the environment, especially in emerging markets.
The methodology described in the paper aligns with the International Standards for Life Cycle Assessment (ISO 14040 and 14044), which enhances the study's credibility. The paper thoroughly describes the system boundaries, functional unit (1 litre of gasoline) and allocation procedures, which ensures transparency and repeatability. The ReCiPe method for impact assessment is appropriate, covering key environmental impact categories such as climate change, soil acidification and human toxicity. Furthermore, including scenario analyses for different gasoline distribution methods adds practical value by providing insights into environmentally beneficial alternatives.
The authors' use of primary data from 2018 reports from government and private organizations enhances the validity of the analysis. Focusing on stable gasoline demand conditions ensures that the data accurately reflects typical operations. Where primary data is unavailable, the authors fill gaps with secondary data from the Ecoinvent database, adding depth and completeness to the LCA.
The scenario analysis in the paper adds a feasibility dimension to the study. By comparing different gasoline transportation methods (rail vs. road), the authors demonstrate potential reductions in environmental impacts, with rail emerging as a particularly sustainable option. The study's findings support strategic decisions about gasoline logistics and can inform policy and infrastructure developments to minimize the environmental footprint of gasoline supply chains.
The paper provides a solid foundation for future comparative research on alternative fuel options in Sri Lanka. The findings highlight the critical role of fuel refining and distribution processes in environmental impacts, which aligns well with previous studies conducted in other regions, contributing to the broader discussion on sustainable fuel supply chains.
Further clarification of the assumptions, especially concerning emissions from imported gasoline, would enhance transparency.
In addition, including an economic cost analysis of the proposed distribution scenarios could enhance the practical implications of the paper.
This paper is valuable to the sustainability literature, especially for regions investigating sustainable fuel supply chains. The authors effectively use life-cycle analysis (LCA) to provide practical information on the environmental impacts of gasoline distribution in Sri Lanka. The paper is well-structured, methodologically sound, and suitable for publication.
Reviewer 3 Report
Comments and Suggestions for Authors
The research in the manuscript is of interest. Their usefulness is reflected by the impact on the environment quantified by the authors. However, the practical application is elliptical. The authors do not justify the choice of the four scenarios. Also, a chapter seems to be missing from the manuscript. After the presentation of the four scenarios, the obtained results are presented. The authors must remove this shortcoming through a broad and practical description of the modeling process, the primary data, and how the results were obtained. Also, in the conclusions chapter, I do not understand the usefulness of abbreviations for terms used only twice in the manuscript. I am waiting for the manuscript's improvement for a second revision round.
Round 2
Reviewer 3 Report
Comments and Suggestions for Authors
The authors have improved the manuscript in the areas I have identified. They have also correctly justified the use of certain terms. Consequently, I recommend publication of the manuscript.